# Employees' Perceptions of Green Supply-Chain Management, Corporate Social Responsibility, and Sustainability in Organizations: Mediating Effect of Reflective Moral Attentiveness

Yasir Hayat Mughal [1,*][ID], Kesavan Sreekantan Nair [1], Muhammad Arif [1][ID], Fahad Albejaidi [1], Ramayah Thurasamy [2,3,4,5,6,7][ID], Muhammad Asif Chuadhry [8] and Saqib Yaqoob Malik [9]

1   Department of Health Administration, College of Public Health & Health Informatics, Qassim University, Albukayriyah 52531, Saudi Arabia; k.nair@qu.edu.sa or ksnair2005@gmail.com (K.S.N.); ma.hajj@qu.edu.sa (M.A.); f.alonazy@qu.edu.sa (F.A.)
2   School of Management, Universiti Sains Malaysia, Minden 11800, Malaysia; ramayah@usm.my
3   Department of Information Technology & Management, Daffodil International University (DIU), Birulia 1216, Bangladesh
4   University Center for Research & Development (UCRD), Chandigarh University (CU), Chandigarh 160036, India
5   Fakulti Ekonomidan Pengurusan (FEP), Universiti Kebangsaan Malaysia (UKM), Bangi 43600, Malaysia
6   Faculty of Economics and Business, Universitas Indonesia (UI), Depok 16424, Indonesia
7   Azman Hashim International Business School, Universiti Teknologi Malaysia (UTM), Kuala Lumpur 54100, Malaysia
8   Department of Education & Leadership, Shifa Tameer-e-Millat University, Islamabad 44000, Pakistan; asif.epm@gmail.com
9   Department of Management Sciences, Preston University, Islamabad 44000, Pakistan; malikhashir58@yahoo.com
*   Correspondence: y.hayat@qu.edu.sa or hbl_rulz@yahoo.com

**Abstract:** (1) Background: The increasing level of concern over reduction non-renewable resources, global warming, pollution, and social issues has led firms to initiate green and social activities. Furthermore, there is limited empirical evidence on the potential impact of green initiatives, corporate social responsibility (CSR), and reflective moral attentiveness (RMA) on sustainable performance. The purpose of this study was to investigate the mediating effect of reflective moral attentiveness (RMA) on the relationship between green supply-chain-management practices (GSCM) and CSR on sustainable performance. Based on the natural-resource-based view, stakeholder resource-based view, and signaling theory, this study investigated the role of GSCM practices and CSR in sustainable performance using cross-sectional data from the manufacturing and services industries from Khyber Pakhtunkhwa province, in Pakistan. (2) Methods: Using a non-probability convenience-sampling method, 500 employees were selected from the firms which are listed on the Pakistan stock exchange (PSE) and questionnaires were distributed. Complete questionnaires were received from 380 employees and used in the analysis, yielding a response rate of 76%. Partial least squares structural equation modeling (PLS-SEM) software was used for the confirmatory-factor analysis (CFA) and the testing of the hypotheses. The CFA results revealed the reliability and validity of the questionnaires. (3) Results: The results of the structural model (hypotheses testing) show that four attributes of GSCM practices (internal environmental management, green purchasing, cooperation with customers, and eco-design) have a positive influence on sustainable performance, while investment recovery and CSR were found to be insignificant. Moreover, there were significant and positive influences of GSCM and RMA on sustainable performance. On the other hand, control variables, such as gender, experience, and age, were found to have no significant role in sustainable performance. A further analysis revealed that reflective moral attentiveness mediated the relationship between GSCM, CSR, and sustainable performance. (4) Conclusions/implications: This study has several implications for green services and manufacturing firms specifically and for practitioners, researchers and academics in general. The innovation and novelty of this study lie in its determination of the contribution of RMA, GSCM, and CSR to achieving sustainable performance. Firms can improve their clean production activities by

incorporating this model as a strategy. Future studies may add moderators and mediators to explore the impact of CSR and GSCM practices upon sustainability.

**Keywords:** green supply-chain-management practices; corporate social responsibility; reflective moral attentiveness; sustainable performance; hospitality and manufacturing industries

## 1. Introduction

Business organizations considered the world as a limitless commodity or good. In addition, these organizations thought that their operations had a very minor influence upon the environment, which led to a reduction in natural resources and an increase in environmental issues. Increases in pollution and environmental problems subsequently pushed organizations to take serious actions to overcome these environmental issues. These environmental issues gave birth to the concept of sustainable performance. Sustainable performance can be divided into three factors: social, economic, and environmental [1]. Increasing rates of global warming and pollution and decreases in non-renewable resources have driven manufacturing firms to initiate green supply-management activities, such as green supply-chain management (GSCM), to obtain competitive advantages and ensure sustainable performance [2]. The extension of the concept of GSCM across the supply chain creates effective innovation in the manufacturing industries [2]. Green supply-chain management (GSCM) has been considered as the most effective strategy for creating eco-sustainability, in order to increase sustainable performance and reduce environmental issues [3]. Furthermore, GSCM is integrated into environmental thinking and supply-chain management, ranging from product design, material sources and their selection, manufacturing processes, and the delivery of products and their end-of-life management [4]. There are studies that highlight the gaps prevailing in the literature on supply chains, corporate social responsibility, and sustainability issues, which were overlooked in the past studies [5–7].

During the last few decades, corporate social responsibility (CSR) has become of great interest to researchers. Due to rapid changes in market conditions and technology, organizations have focused on their stakeholders' concerns regarding the environment, sustainability, and social responsibility [5,8]. In addition, CSR is considered three different perspectives: internal, external and environmental responsibilities [5]. The internal responsibilities of firms include the safety, health, and well-being of employees, increasing the standard and quality of their lives, on-the-job training and development, work–life balance, and appropriate working environments. The external responsibilities of firms include identifying the social problems of societies and communities and providing solutions to these issues. The environmental responsibilities of the firms include reducing the wastage of natural resources, reducing emissions of carbon-dioxide gases ($CO_2$), reducing the wastage of water, energy and power, reductions in the consumption of paper, and the preservation of natural resources [5].

The current research shows that GSCM practices and CSR are directly linked to the sustainable performance of firms [8]. However, some researchers have identified several linking mechanisms between GSCM, CSR, and sustainable performance, which include organizational-level mediators, such as green climate, and individual-level mechanisms, such as employee environmental commitment [9]. According to Garavan et al. [10], few studies have investigated the socio-cognitive characteristics of employees as potential linking mechanisms. Therefore, there is a need to investigate socio-cognitive characteristics as linking and mediating mechanisms in order to establish the association between GSCM, CSR, and sustainable performance.

This study aims to investigate the mediating effect of reflective moral attentiveness on the relationship between GSCM, CSR, and sustainable performance. A relatively new concept, RMA explains 'the extent to which an individual chronically perceives and considers

morality and moral elements in his or her experience' [9]. This study predicts that signals communicated through GSCM practices and CSR will gain the attention of stakeholders and will then be reflected in the competitive advantage and sustainable performance they ensure.

A very important finding that emerged from the data in this study is that, with the exception if investment recovery and CSR, all the remaining GSCM practices are potentially effective in signaling to employees the importance of sustainability. Additionally, RMA acts as a mediating variable between CSR, GSCM, and sustainable performance.

This paper is structured as follows. Section 1 presents the introduction and the aim of the study. Section 2 presents relevant research and the study's hypotheses. Section 3 includes the research methodology, study population, sampling and data-collection tools and techniques, analytical strategy, control variables, measurement model, and interpretation. Section 4 presents the study's findings and the structural model. Section 5 presents a discussion of the main findings in the context of the available research. Section 6 presents theoretical contributions. Section 7 includes the practical implications of the study. Section 8 outlines the conclusions and recommendations. Section 9 features the limitations of this study and directions for further research.

## 2. Literature Review

### 2.1. Sustainable Performance

The concept of sustainability was first introduced by the World Commission on Environment and Development (WCED) in 1987 [11]. Since then, it has become a topic of interest for practitioners, academics, and researchers. The fourth industrial revolution and frequent changes in industries have created several environmental issues, such as air and water pollution, which firms are now interested in handling and controlling. The WCED defines sustainability as "development that meets the present needs without compromising the future generation's requirements" [11].

Subsequently, Elkington [12] called sustainable performance a triple-bottom-line principle for sustainable performance. It has three attributes, i.e., economic performance, environmental performance, and social performance. Economic performance concerns the financial matters and performance of firms, environmental performance is related to reducing environmental issues and wastage of resources, and social performance concerns the welfare of employees, societies, communities, creditors, suppliers, and other stakeholders. In the literature, more attention is given to economic and environmental performances than social performance, which raises awareness of social issues [2].

Based on the discussion above, this study applies the triple-bottom-line principle. According to Yusliza et al. [1], the initiation of green objectives and their alignment with firms' objectives would help firms to attain competitive advantages and sustainable performance. Therefore, the socially responsible behavior of firms is as important as their economic and environmental behavior. According to Azam et al. [13], climate changes, pressure from stakeholders, and global warming are the issues due to which sustainability is receiving interest from firms and explain why their senior managers are considering the importance and significance of sustainability. Sustainability does not only include financial gains; it also involves tackling environmental issues and taking care of employees, as well as ensuring the well-being of societies and other stakeholders [14].

### 2.2. Green Supply-Chain-Management Practices

The concept of green supply-chain management was introduced by Green et al. [15] to develop and build environmentally friendly management practices in supply chains. Srivastava [16] states that green supply-chain-management practices include environmental thinking in SCM. This includes different stages, i.e., product design, material sources and selection, manufacturing processes, product delivery, and the end-of-life management of product. According to Srivastava, initial studies only focused on green purchasing and reverse logistic; however, subsequently, other researchers also studied green supply-chain

management and its environmental scope [2,17]. Although the majority of the studies conducted on green supply-chain-management practices used GRSM, which has extensive practical applications, no comprehensive and holistic framework has been established for these practices at the time of writing [18].

Different authors have varying opinions about green supply-chain-management practices and their dimensions. For example, Srivastava [16] claimed that green design, green purchasing, production, distribution, logistics, marketing, and reverse logistics are the dimensions of GSCM. However, Luthra et al. [19] are of the view that all the product-life-cycle phases are included in GSCM, starting from the extraction of the raw material to the end of the product's life cycle. Furthermore, Srivastava [16] contends that the boundaries and limits of GSCM are dependent upon the objectives of the researcher.

In view of observations above, in this study, green supply-chain-management practices and CSR are used as predictors, whereas sustainable performance is used as a criterion. According to Mardani et al. [17], GSCM has five attributes, namely, internal environment management, green purchasing, cooperation with customers, co-design, and investment recovery, which are explained below with reference to sustainable performance.

### 2.2.1. Internal Environment Management (IEM) and Sustainable Performance

Internal environment management concerns the ways in which a firm's own policies are developed and implemented to support the environment [14,20]. Activities conducted by upper- and mid-level management to support environmental practices, inter-departmental cooperation for environmental protection, and the implementation of a system that helps to save the environment, are the activities that constitute the internal environment management [21]. The practice of GSCM methods helps organizations to enhance their sustainable performance. In order to reduce environmental issues, firms need to better understand and highlight issues related to the environment, such as production and transportation. According to Vanalle et al. [22], during the process of production, several issues, such as waste, air, water, and soil pollution, affect the environment. Therefore, it is crucial to control these issues. Firms can decrease the negative effects of the production process on the environment by implementing GSCM practices. These practices can positively influence environmental performance and community health by minimizing the utilization of natural resources [23]. Based on the discussion above, the following hypotheses are postulated.

**Hypothesis 1 (H1).** *Internal environment management has a positive effect on sustainable performance.*

### 2.2.2. Green Purchasing and Sustainable Performance

According to Kirchoff et al. [24], the integration of environmental issues into procurement process is called green purchasing. This is the first step in a value-chain process. The success of green purchasing mainly depends upon the environmental objectives of firms. This is why green purchasing is considered the most important component and dimension of GSCM practices [25]. The selection of the right supplier has a significant role in accomplishing the environmental objectives of firms; however, the selection of correct supplier is insufficient. Firms must have a collaborative understanding with their suppliers and they must ensure that the selected supplier meets the environmental criteria they set [26]. Green purchasing has a positive and significant role in attaining competitive advantage through sustainable performance [27]. Thus, this study proposes the following hypothesis.

**Hypothesis 2 (H2).** *Green purchasing has a positive effect on sustainable performance.*

### 2.2.3. Cooperation with Customers and Sustainable Performance

According to Green et al. [15], firms adopt GSCM practices because of pressure from their stakeholders. The creation of green supply chains and cooperation with customers and stakeholders are effective strategies to enhance economic, environmental, and social

performance. From the discussion above, it is concluded that cooperation with customers helps firms to gain a competitive advantage and attain sustainable performance. Therefore, the following hypothesis is proposed:

**Hypothesis 3 (H3).** *Cooperation with customers has a positive effect on sustainable performance.*

2.2.4. Eco-Design and Sustainable Performance

Eco-design is also known as green design. Eco-design minimizes the negative effects of production on the environment [28]. Eco-design practices include product design, the ideal consumption of energy, raw materials, the reduced use of hazardous materials, and the generation of waste during product development [29]. Eco-design allows firms to use solar and biodegradable energy sources to reduce environmental effects and enhance performance [30]. According to Zhu et al. [27] eco-design gives access to green markets and remanufacturing, which leads firms towards sustainability. This positive role of eco-design in sustainable performance is also reported in other studies [27,29]. Thus, the following hypothesis is postulated.

**Hypothesis 4 (H4).** *Eco-design has a positive effect on sustainable performance.*

2.2.5. Investment Recovery and Sustainable Performance

Investment recovery is a traditional practice of firms, in which excessive scrap materials, inventories, and used materials are resold [31]. The aim of investment recovery is to recover the value of obsolete and surplus items. Firms include these items in reverse logistical processes so that they may be disposed of correctly [31]. Reverse logistics also include product return, reuse, and recycling, the recollecting of products from customers, and their reuse. They are positively related to environmental performance. Therefore, firms cannot achieve sustainability without the appropriate management of reverse logistical processes [27,31]. Thus, the following hypotheses are developed.

**Hypothesis 5 (H5).** *Investment recovery has a positive effect on sustainable performance.*

**Hypothesis 6 (H6).** *GSCM has a positive effect on reflective moral attentiveness.*

*2.3. Corporate Social Responsibility and Sustainable Performance*

Corporate social responsibility leads to social, financial, environmental, and economic improvements in performance [32]. Through CSR, firms can create a positive image of their business in the eyes of customers and societies and may obtain competitive advantages and sustainable performance as a result. Mughal et al. [33] has also reported a positive relationship between CSR and financial performance. Based on the discussion above, the following hypotheses are proposed.

**Hypothesis 7 (H7).** *Corporate social responsibility has a positive effect on sustainable performance.*

**Hypothesis 8 (H8).** *Corporate social responsibility has a positive effect on reflective moral attentiveness.*

*2.4. Mediating Role of Reflective Moral Attentiveness in GSCM, CSR, and Sustainable Performance*

According to the signaling theory [10], employees ignore some environmental signals and focus on others. This process depends on the extent to which these signals are cognitively accessible and important. Employees focus on the moral and ethical aspects of the environmental and green signals communicated to them through GSCM and CSR. These employees assess these signals through their moral lens [10].

Reflective moral attentiveness is "the intentional reflection of employees regarding environmental issues," which plays a major and crucial role as a mediating mechanism between GSCM, CSR, and sustainable performance [9,10]. Therefore, it is hypothesized that when employees are more attentive morally, they are more likely to evaluate signals.

In view of the discussion in the literature review, we propose that RMA is linked to enhance sustainable performance, social responsibility behaviors, green behaviors, ethical decision making, and reduced unethical behavior [34].

**Hypothesis 9 (H9).** *Reflective moral attentiveness has a positive effect on sustainable performance.*

**Hypothesis 10 (H10).** *Reflective moral attentiveness mediates the positive relationship between green supply-chain-management practices and sustainable performance.*

**Hypothesis 11 (H11).** *Reflective moral attentiveness mediates the positive relationship between corporate social responsibility and sustainable performance.*

*2.5. Theoretical Framework*

An eclectic theoretical approach was used to inform this study, as no single theory is able to explain the relationship between GSCM, CSR, RMA, and sustainable performance. Barney [35] was the first to introduce the concept of the resource-based view (RBV). The resource-based view is used globally by researchers to report the impact of green activities, such as green supply-chain-management practices (GSCM), on firm performance. In addition, Barney [35] claimed that internal resources help firms to gain competitive advantages and sustainable performance. Subsequently, Jabbour [36] further explained that there are two types of resource, i.e., tangible resources, such as buildings, equipment, and machines, and intangible resources, such as leadership, intellectual capital, skills, creativity, and positive social reputation. According to Hart [37], tangible resources provide temporary competitive advantages to firms because these types of resource can easily be imitated by competitors, whereas intangible resources are difficult to imitate as they are gained by experience. Hart [37] noted that firms' competitive advantages and sustainable performance are threatened by natural environmental phenomena, such as the degradation of the eco-system and the reduction in natural resources. The scope of the RBV was extended, and a new typology, the natural-resource-based view (NRBV), was introduced by Hart. According to this new approach, firms can attain competitive advantages by implanting strategies such as the prevention of pollution, reducing the waste of water and natural resources, and sustainable development.

According to Cankaya and Sezen [14], green supply-chain-management practices (GSCM) can be considered as strategic resources to attain sustainable performance through the lens of NRBV. These authors further argue that GSCM practices cannot be easily imitated by competitors as these are based on experience, knowledge, and skills. For example, it takes several years for a firm to develop a positive social image, which cannot be easily imitated by its competitors. Similarly, Wiejithilake [38] argues that firms with green initiatives have more positive images, increased sales, and enhanced sustainable performance. Therefore, firms with green activities can reduce costs, increase their capabilities, production, and environmental performance, reduce waste, effectively manage natural resources, and improve quality, leading to more social acceptance.

The stakeholder theory, introduced by Freeman [39], is widely used to explain the effects of CSR on firms' sustainable performances. Previously, firms were product-oriented and focused only on economic performance, i.e., profitability. However, due to increases in competition, environmental damages have led to an increase inattention to social responsibility. As social responsibility gained importance, the concept of the stakeholder became prominent. Freeman [39] divided stakeholders into two groups, i.e., internal stakeholders (managers, employees, and owners), and external stakeholders (creditors, society, customers, clients, government, competitors, and suppliers). The stakeholder theory argues that firms should meet the expectations of their stakeholders in an efficient and effective way. Stakeholders with an awareness of the environment and green activities desire business firms that behave ethically and show positive attitudes towards both environmental and social issues, along with economic performance. Therefore, it is essential for firms to initiate green practices to establish better relationships with stakeholders. Based on

the discussion above, an eclectic theoretical approach was used to investigate the roles of GSCM practices, CSR, and RMA in sustainable performance (Figure 1).

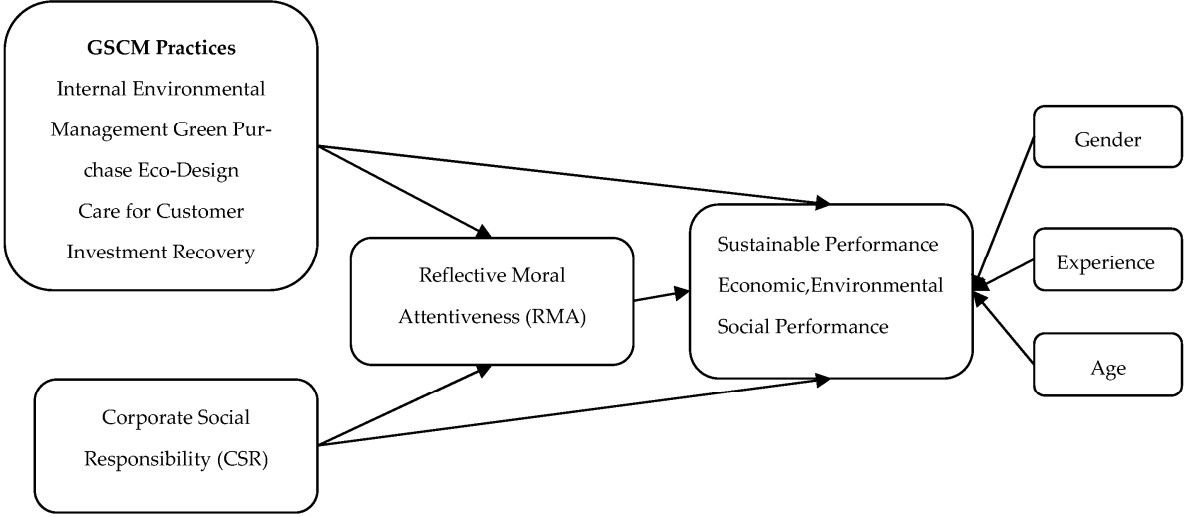

**Figure 1.** Theoretical framework.

## 3. Materials and Methods

### 3.1. Population, Sampling, and Data-Collection Methods

This was a quantitative study. Survey approach was used in this study. The nature of the data was cross-sectional, i.e., data were collected at one point intime. The primary data were collected during 2022 and 2023 through self-administered questionnaire.

The population of this study comprised employees inhospitality and manufacturing industries in Khyber Pakhtunkhwa province, Pakistan. The industries included, cement, sugar, hospitality, leather, furniture, textiles, agriculture, food and beverages, dairy, and plastic. Manufacturing and hospitality firms were chosen because they are most closely related to environmental issues. More than 3 million firms are registered, of which approximately 19.72% are manufacturing firms while 22.3% are hospitality firms, hotels, and restaurants. Non-probability convenience-sampling technique was used to select the sample size. The firms which were listed on Pakistan stock exchange were selected for the current study. The reason for these criteria is that their employees had knowledge and awareness about benefits of green activities and were able to answer all the items on the questionnaire easily.

In total, 500 employees from manufacturing and hospitality firms were selected and, after obtaining permission from relevant departments, questionnaires were distributed to relevant employees. Respondents were assured that this study was for academic purposes and data and identity of the respondents would be kept confidential. Questionnaires were sent via post to respondents who lived far away and return envelope were attached with specific codes to avoid duplication. In total, 380 completed questionnaires were received by researchers, yielding a response rate of 76%.

From Table 1, it is evident that majority of participants in the study were male, totaling 344 (90.5%), while 36 (9.5%) were female. Further, respondents were asked about their age, and it was found that majority of the respondents were aged 46–55 years (140 (36.8%)), followed by the age group of 56 and above (118 (31.15)). Sixty-seven (17.6%) respondents belonged to age group of 36–40 years, and 49 (12.9%) were 31–35 years old, while only 6 (1.65) were 25–30 years of age. Moreover, regarding education, majority of the respondents were educated to Master's level 286 (75.3%), followed by those with undergraduate education 71 (18.7%); in total, 21 (5.52%) respondents had doctoral degrees and only 2 (0.55) had diploma-level education. Respondents were also asked about experience and length of service. It was revealed by the results that 137 (36.1%) respondents had 6–10 years of expe-

rience, followed by those with 11–15 years of experience (99 (26.1%)). Seventy-two (18.9%) respondents had 16–20 years of experience and 59 (15.55) had 1–5 years of experience, while only 13 (3.45) had more than 20 years of experience. A further analysis of data results indicated that employees who participated in the survey from sugar and cement industry numbered 95 (25%), while hospitality, tourism, and leisure industry accounted for 57 (15%), food and beverages for 68 (18%), leather for 46 (12%), textiles for 19 (5%), agriculture and fruit processing for 53 (14%), furniturefor27 (7%), and construction industry for 15 (4%).

**Table 1.** Sample Characteristics.

| Variables | *n* | % |
|---|---|---|
| Male | 344 | 90.5 |
| Female | 36 | 9.5 |
| 25 to 30 years of age | 6 | 1.6 |
| 31–35 years of age | 49 | 12.9 |
| 36–40 years of age | 67 | 17.6 |
| 46–55 years of age | 140 | 36.8 |
| 56 and above | 118 | 31.1 |
| Diploma education | 2 | 0.5 |
| Undergraduate education | 71 | 18.7 |
| Master's education | 286 | 75.3 |
| Doctoral education | 21 | 5.52 |
| 1–5 years of experience | 59 | 15.5 |
| 6–10 years of experience | 137 | 36.1 |
| 11–15 years of experience | 99 | 26.1 |
| 16–20 years of experience | 72 | 18.9 |
| Over 20 years of experience | 13 | 3.4 |
| Industry Type | | |
| Sugar and cement | 95 | 25 |
| Hospitality, tourism, and leisure | 57 | 15 |
| Food and beverages | 68 | 18 |
| Leather | 46 | 12 |
| Textiles | 19 | 5 |
| Agriculture and fruit processing | 53 | 14 |
| Furniture | 27 | 7 |
| Construction | 15 | 4 |

*3.2. Measurement*

All the validated questionnaires were adopted from previous studies. For instance, sustainable-performance instrument was validated in several studies, such as those by Yusliza et al. [1] and Yong et al. [40]. It is a seven-point Likert scale ranging from 1 = not at all to 7 = to great extent, with three constructs: economic performance (5 items), environmental performance (5 items), and social performance (5 items). In total, the questionnaire has 15 items. Green supply-chain management scale, adopted from Micheli et al. [2], is a 20-item scale, with seven-point scale ranging from 1 = not considering to 7 = implementing successfully. Corporate social responsibility scale, adopted form Abbas et al. [5], it's a four-item scale ranging from 1 = strongly disagree to 7 = strongly agree. Reflective moral attentiveness questionnaire was adopted from Garavan et al. [10]. It has six items. After

CFA, one item of CSR, one item of RMA, and one item of IR were excluded due to low factor loadings. Details of all variables and respective items are given in Table 2.

**Table 2.** Summary of the variables.

| S# | Variables | No. of Items | Questioners Sources |
|---|---|---|---|
| 1 | Internal environment management (IEM) | 6 | Micheli et al., 2020; Azam et al., 2022; Cankaya and Sezen, 2019 [2,13,14] |
| 2 | Green purchase (GP) | 5 | Micheli et al., 2020; Azam et al., 2022; Cankaya and Sezen, 2019 [2,13,14] |
| 3 | Eco-design (ECO-D) | 3 | Micheli et al., 2020; Azam et al., 2022 [2,13] |
| 4 | Care/cooperation with customers (CC) | 3 | Micheli et al., 2020; Azam et al., 2022 [2,13] |
| 5 | Investment recovery (IR) | 2 | Micheli et al., 2020; Azam et al., 2022; Cankaya and Sezen, 2019 [2,13,14] |
| 6 | Corporate social responsibility (CSR) | 3 | Abbas et al., 2019 [5] |
| 7 | Reflective moral attentiveness (RMA) | 5 | Garavan et al., 2022 [10] |
| 8 | Sustainable performance (SP) Economic performance (ECP), environmental performance (ENP) & Social performance (SCP) | 15 | Yong et al., 2019; [40] |

### 3.3. Data-Analysis Techniques

We used Smart PLS-SEM software, a form of second-generation structural-equation-modeling software that can test the measurement model and structural model simultaneously [41]. According to Ramayah et al. [42] and Hair et al. [43], this software is suitable for testing complex structural-equation models. The PLS-SEM is a second-generation statistical software/package for analyzing complex and difficult models. This software is one of the most advanced tools for helping researchers to obtain sophisticated statistical results. Smart PLS is suitable for non-normal as well as small data sets. In the first step, researchers have to develop a measurement model in which confirmatory factor analysis (CFA) needs to be run. In CFA factor loadings >0.70, AVE > 0.5, CR > 0.7, and alpha value > 0.70 are investigated. If an item is found to be problematic and does not meet the threshold, then it maybe excluded to obtain the model with the best fit. In CFA, convergent validity (AVE and CR) is investigated, followed by discriminant validity. In addition, once researchers find the model with the best fit in CFA, they can proceed to structural model, in which hypotheses can be tested through bootstrapping, t-statistics, lower- and upper-limit confidence intervals, and significance values.

### 3.4. Control Variables

We included three control variables, age, gender, and experience in the research model. Past studies reported that gender, age, and experience play significant roles in predicting green behavior and obtaining sustainability [44]. As part of our analysis, we checked the impact of age, gender, and experience on sustainable performance and found that all control variables had insignificant influences on sustainable performance.

### 3.5. Measurement Model

Hair et al. [43] provided the guidelines we used to assess the measurement model by assessing first the loadings (≥0.7) (Appendix A), average variance extracted (≥0.50), and composite reliability (≥0.70) (see Table 3). Since we had a second-order construct (bolded),

Sustainable Performance (3 dimensions, italics), we assessed the validity and reliability of the first-order constructs as well as assessing the validity and reliability of the second-order constructs. Thus, based on the values presented in Table 3, we can conclude that we had sufficient convergent validity and reliability (Figure A1, Appendix B). Moreover, highest mean value was scored by investment recovery, M = 6.468, S.D = 0.998, followed by eco-design, M = 6.299, S.D = 1.030, and environmental performance M = 6.159, S.D = 1.128. Care for customers, social performance, and sustainable performance also displayed mean values higher than 6, while remaining constructs scored mean values between 5.4 and 5.7. To assess discriminant validity, we used the HTMT ratio, as suggested by Franke and Sarstedt [45] and Cho Chung, Young [46]. According to the guidelines, if the HTMT ratios are ≤0.85 or <1 [43], then we can conclude that discriminant validity has been achieved. As shown in Table 4, all the HTMT ratios met the threshold; thus the measures in our study had good discriminant validity.

**Table 3.** Descriptions and quality of measurement items.

| Constructs | Mean | Std Dev. | Kurtosis | Skewness | CR | AVE |
|---|---|---|---|---|---|---|
| CC | 6.049 | 1.288 | −0.139 | −1.005 | 0.949 | 0.861 |
| CSR | 5.732 | 1.582 | −0.472 | −0.982 | 0.970 | 0.915 |
| ECO-D | 6.299 | 1.030 | 1.096 | −1.077 | 0.945 | 0.851 |
| GP | 5.774 | 1.653 | −0.663 | −0.986 | 0.983 | 0.921 |
| IEM | 5.740 | 1.659 | −0.579 | −1.017 | 0.984 | 0.914 |
| IR | 6.468 | 0.998 | 1.141 | −1.137 | 0.985 | 0.970 |
| RMA | 5.405 | 0.952 | 3.347 | −1.771 | 0.896 | 0.633 |
| SP | 6.013 | 1.210 | −1.176 | −0.717 | 0.987 | 0.821 |
| *ENP* | *6.159* | *1.128* | *0.045* | *−1.084* | *0.970* | *0.868* |
| *ECP* | *5.758* | *1.509* | *−1.078* | *−0.733* | *0.975* | *0.885* |
| *SCP* | *6.067* | *1.213* | *−0.776* | *−0.642* | *0.965* | *0.847* |

**Table 4.** Discriminant validity.

| Variables | 1 | 2 | 3 | 4 | 5 | 6 | 7 | 8 | 9 |
|---|---|---|---|---|---|---|---|---|---|
| 1. CC | | | | | | | | | |
| 2. CSR | 0.837 | | | | | | | | |
| 3. ECOD | 0.888 | 0.731 | | | | | | | |
| 4. ECP | 0.928 | 0.931 | 0.773 | | | | | | |
| 5. ENP | 0.948 | 0.917 | 0.856 | 0.990 | | | | | |
| 6. GP | 0.924 | 0.941 | 0.765 | 0.966 | 0.966 | | | | |
| 7. IEM | 0.881 | 0.973 | 0.729 | 0.964 | 0.959 | 0.986 | | | |
| 8. IR | 0.879 | 0.939 | 0.780 | 0.945 | 0.970 | 0.955 | 0.953 | | |
| 9. RMA | 0.048 | 0.113 | 0.030 | 0.063 | 0.056 | 0.079 | 0.084 | 0.064 | |
| 10. SCP | 0.943 | 0.937 | 0.799 | 0.996 | 0.975 | 0.972 | 0.971 | 0.948 | 0.051 |

## 4. Structural Model

To test the hypotheses, we developed the structural model in PLS-SEM, and we ran a bootstrapping procedure with 5000 resamples to generate the standard errors, t-values, *p*-values, and bias-corrected confidence intervals [43]. First, we assessed the variance explained in our model. The $R^2$ was 0.957 ($Q^2$ = 0.845), indicating that the model can explain 95.7% of the variance in Organizational Performance, and the predictive relevance $Q^2$ was 0.845, which was greater than 0, indicating the sufficient predictive relevance of our model. Internal Environmental Management (β = 0.407, $p < 0.01$), Green Purchasing (β = 0.147, $p < 0.05$), Cooperation with Customers (β = 0.283, $p < 0.01$), and Eco-Design (β = 0.140, $p < 0.01$) were positively related to Sustainable Performance, while Investment Recovery and Corporate Social Responsibility were not significant. These results supported H1, H2, H3, and H4, while H5 was not supported. The GSCM had a positive influence on the RMA (β = 0.221, $p < 0.01$), supporting H6; the CSR had a positive influence on the SP (β = 0.023, $p > 0.05$), so H7 was not supported; the CSR had a positive influence on the

RMA (β = 0.666, *p* < 0.01), so H8 was supported; and the RMA had a positive influence on the SP (β = 0.852, *p* < 0.01), so H9 was supported (see Table 5). A further analysis of the indirect effects revealed that the RMA mediated between the GSCM and the SP (β = 0.188, *p* < 0.01) and between the CSR and the SP (β = 0.568, *p* < 0.01).Thus, H10 and H11 were also substantiated. The most important and dominant predictor of Sustainable Performance was the effect of RMA on SP because this showed highest beta value. These findings highlight the important role of reflective moral attentiveness (see Tables 5 and 6).

**Table 5.** Hypotheses testing.

| Hypothesis | Relationship | Std Beta | S.E | t-Value | *p*-Value | BCI LL | BCI UL | Support |
|---|---|---|---|---|---|---|---|---|
| H1 | IEM→SP | 0.407 | 0.093 | 4.380 | 0.000 | 0.259 | 0.566 | Yes |
| H2 | GP→SP | 0.147 | 0.086 | 1.723 | 0.043 | 0.013 | 0.294 | Yes |
| H3 | CC→SP | 0.283 | 0.055 | 5.168 | 0.000 | 0.207 | 0.376 | Yes |
| H4 | ECOD→SP | 0.140 | 0.026 | 5.378 | 0.000 | 0.093 | 0.178 | Yes |
| H5 | IR→SP | 0.040 | 0.025 | 1.605 | 0.055 | −0.004 | 0.075 | No |
| H6 | GSCM→RMA | 0.221 | 0.05 | 4.427 | 0.000 | 0.128 | 0.137 | Yes |
| H7 | CSR (SP) | 0.023 | 0.050 | 0.453 | 0.325 | −0.064 | 0.102 | No |
| H8 | CSR (RMA) | 0.666 | 0.047 | 14.133 | 0.000 | 0.578 | 0.75 | Yes |
| H9 | RMA (SP) | 0.852 | 0.018 | 47.97 | 0.000 | 0.817 | 0.886 | Yes |
|  | Age (SP) | 0.063 | 0.045 | 1.40 | 0.161 | −0.02 | 0.159 | No |
| Control | Gender (SP) | 0.02 | 0.021 | 0.006 | 0.32 | −0.02 | 0.059 | No |
| Variables | Experience (SP) | −0.091 | 0.049 | 1.833 | 0.067 | −0.2 | 0.002 | No |

**Table 6.** Indirect effects (mediation effects).

| Indirect Relationships | Std Beta | S.E | T | P | BCI LL | BCI UL | Support |
|---|---|---|---|---|---|---|---|
| H10 GSCM→RMA→SP | 0.188 | 0.043 | 4.344 | 0.000 | 0.11 | 0.274 | Yes |
| H11 CSR→RMA→SP | 0.568 | 0.042 | 13.671 | 0.000 | 0.494 | 0.649 | Yes |

## 5. Discussion

Based on the signaling theory, NRBV, and stakeholder theory, the current study hypothesized that GSCM practices and CSR would have a positive influence on sustainable organizational performance with the mediating effect of reflective moral attentiveness. The results revealed that only four dimensions of GSCM practices, i.e., internal environment management, green purchasing, cooperation with customers, and eco-design, have a positive and significant effect on sustainable performance, while investment recovery and CSR have insignificant effects on sustainable performance. The majority of the studies conducted in this area supported this argument. For example, Cankaya and Sezen [14], Micheli et al. [2], and Schmidt et al. [29] reported positive effects of GSCM practices on sustainable organizational performance. This implies that GSCM practices may provide competitive advantages and ensure sustainable performance.

This study found that internal environment management has a significant effect on the triple bottom line, i.e., sustainable performance (SP). If firms wish to make their supply chains green, first, they need to undertake initiatives in internal environmental management. They need to collaborate and integrate with all the members of the SC. Azam et al. [13] reported that IEM is a successful driver of triple bottom line, i.e., sustainable performance. They also indicated that firms with serious concerns over environmental issues implement environmental initiatives to ensure successful GSCM [13,46]. Moreover, Micheli et al. [2] and Cankaya and Sezen [14] also found a significant role of IEM in sustainability. Therefore, it was proven that IEM plays an important role in sustainable performance in hospitality and manufacturing industries. Thus, H1 was supported.

In terms of the relationship between green purchasing and sustainable organizational performance, this study found that green purchasing has a significant impact on sustainable organizational performance, the purchasing activities of firms, environmental efforts, and

the objectives of firms. The selection of the correct supplier is one of the most important steps in procurement processes. The choice of the correct supplier can have a significant impact on a firm's environmental objectives. Once a firm chooses its suppliers, appropriate strategic and collaborative understanding should be established between these suppliers and the firm. The suppliers must meet the environmental criteria of the firm [2]. By contrast, Zhang et al. [27] and Cankaya and Sezen [14] argued that as green purchasing is an external GSCM dimension, a single firm cannot completely implement this dimension of GSCM. Malik et al. [47] suggested that dominant firms can play a role in the successful implementation of this dimension. Therefore, the role of green purchasing in sustainability was proven. Thus, H2 was supported.

A further analysis of the results revealed that cooperation and collaboration with customers, suppliers, and external stakeholders are considered effective tools for GSCM practices to achieve sustainable performance. Cooperation must be established between all the departments in a firm and its customers and stakeholders to deal with environmental issues and attain competitive advantages. Firms must keep close relationships with their customers and suppliers and support them to implement environmental practices. Firms can learn about the changing trends and demands of customers, as well as the concerns of customers and suppliers over environmental issues, by maintaining close relationships with these customers [48]. Thus, H3 was supported.

This study also found a positive relationship between eco-design and sustainable performance. Eco-design allows firms to use efficient friendly energy resources, such as solar energy and biodegradable sources to reduce environmental issues and enhance sustainable performance. Eco-design practices might help organizations to access green markets. This finding is in line with the results of some studies [2,14], but contradicts those of Esfahbodi et al. [49], who found no relationships between eco-design and economic performance. This might have been due to the fact that, since eco-design activities initially require financial resources and capital, firms consider the cost burdens. Therefore, the role of eco-design in sustainable performance was apparent in this study. Thus, H4 was supported.

Furthermore, H5, i.e., the relationship between investment recovery and sustainable performance, was not supported in this study, as there was no significant relationship between investment recovery and sustainable performance. This finding is in line with the findings of earlier studies that indicated that investment recovery is less attractive in developing economies [49]. Recycling activities require huge investment and, due to the lack of recycling infrastructure, developing economies do not invest in these activities. This could be a reason why the GSCM did not have any significant effects on sustainable performance. According to Geng et al. [28], advances in technology and more investments are required for manufacturing and hospitality firms. Therefore, H5 was not supported. This study also shows that green supply-chain-management practices have significant effects on reflective moral attentiveness. The initiation of green activities in supply chains sends signals to employees to take care of the environment, highlighting the ethical aspect of the supply chain. Furthermore, the RMA shows the extent of employees' ethical and moral values. This finding is also in line with the findings of Garavan [10]. Thus H6 was substantiated.

The current study did not find any relationships between corporate social responsibility (CSR) and sustainable performance. Organizations always aim to attract investors; however, before investing in a firm, investors investigate the firm's annual financial reports and statements. Some investors consider CSR a burden on their profits and do not invest in firms that are actively involved in CSR activities [33,34]. This could be a reason for the insignificant relationship between CSR and SP. Thus, H8 was substantiated. Moreover, this study found that RMA is positively related with sustainable performance. This finding is in line with Sturm's study [50], which found that RMA is responsible for enhancing socially responsible behaviors and helps firms to achieve competitive advantages and enhanced sustainable performance. Thus, H9 was supported. This study also found a mediating effect of RMA, which revealed a positive and significant effect on the relationship between GSCM, CSR, and sustainable performance.

## 6. Theoretical Contributions

This study contributed to the literature on GSCM practices, CSR, and RMA in several ways. First, using the socio-cognitive characteristics of employees' perceptions, such as reflective moral attentiveness, through the lens of signaling theory, this study found that GSCM and CSR influence employees' RMA, which, in turn, affects sustainable performance. This confirms the validity of RMA as a linking mechanism. The application of this construct in a non-Western context through the revelation of its value by linking it to four GSCM practices (except investment recovery) and sustainable performance is relatively new in the literature. It is suggested that when GSCM practices and CSR contain moral and ethical signals, they have significant effects in that they prompt morally attentive employees to develop positive and productive feelings about environmental and green issues. Overall, this finding highlights the importance of the cognition of employees in obtaining sustainable performance. Second, the use of control variables such as age, gender, and experience in this research model showed no significant effects. Third, CSR is indirectly related to sustainable performance through linking mechanisms such as RMA. Since RMA is responsible for enhanced socially responsible behavior [50], the indirect role of RMA in the relationships between CSR and sustainable performance is confirmed. This also confirms the reliability and validity of RMA as a mediating variable, which is anew construct in the literature on GSCM [10].

## 7. Conclusions

In terms of findings and results, internal environment management, green purchasing cooperation with customers, and eco-design has significant influences on sustainable organizational performance. By contrast, investment recovery and corporate social responsibility have insignificant effects on sustainable performance. Furthermore, GSCM and its four dimensions (internal environmental management, eco-design, co-operation with customers, and green purchases) were found to be responsible for enhancing sustainable performance. Investment recovery was not found to be a significant predictor of sustainable performance. This study also shows that CSR is not significantly related to sustainable performance.

Based on the discussion above, it is concluded that green activities and sustainable performance are receiving attention from firms, researchers, and academics. Therefore, firms focus on green supply-chain-management practices to obtain competitive advantages and ensure sustainable performance.

## 8. Practical Implications

The findings of this study provide implications for policy and practice in the hospitality and manufacturing industries. A number of firms in developing economies are interested in avoiding economic risks and in improving their economic situation. However, it is not possible for firms to prefer short-term benefits and expect to survive in the long term by ignoring environmental and social issues. This study highlights the significance and importance of maximizing profits, but firms must also consider social benefits. To this end, managers must conduct cost–benefit analyses. With the implementation of GSCM practices, some costs increase, including investment, procurement, and training costs, while other costs reduce. Firms need to establish close relationships with their suppliers and provide them support in environmental practices.

Another aspect of GSCM practices that has received limited attention from manufacturing and hospitality firms is investment recovery. This limitation is due to the fact that investment recovery requires initial capital to initiate recycling activities. Once investment recovery is initiated, it helps to reduce overall costs by reducing the wastage of materials and energy. Managers can increase the economic performances of their firms by considering the advantages of green activities and reducing the negative impact of production processes on the environment.

Furthermore, CSR activities are not sources of expenditure, but rather sources of investment and management strategies to enhance performance and obtain competitive advantages [46]. Managers and firms are encouraged to invest in community services through donations, social welfare support, free education, and the provision of free food, especially during crises and emergency situations. Firms should increase investment in CSR activities to help people during crises and emergencies. Investment in social activities under CSR helps firms to attract more investors and customers and to enhance their economic performance. Furthermore, CSR creates positive images of firms in the eyes of other stakeholders.

Corporate social responsibility has several benefits for firms, such as the retention of talented staff, who are made to feel important through their membership of ethical firms, leading to increases in firm performance. Managers might establish close and better relationships with creditors, suppliers, and other stakeholders to obtain competitive advantages and attain sustainable performance by controlling environmental issues. This increases the sale of their products and services through CSR. Furthermore, CSR helps these firms to raise the living standards of societies.

## 9. Limitations and Future Research Directions

This study focused on the manufacturing and hospitality industries; therefore, its findings are only generalizable to these industries. Future studies may include retailers and wholesalers. Furthermore, this study analyzed cross-sectional data using PLS-SEM software. Future studies may use longitudinal and qualitative data to obtain a more in-depth understanding of this phenomenon. It is recommended that green educations are included in future frameworks.

This study investigated the indirect impact of the role of RMA in CSR and GSCM practices on sustainable performance. Future studies may be conducted on moderators such as environmental dynamism, social control, supply-chain partners, and supplier involvement, as well as mediators such as green intellectual capital. Similarly, CSR was used as a predictor in this study; it is suggested that further studies focus on corporate citizenship, stakeholder management, business ethics, the creation of shared values, and company values.

**Author Contributions:** Conceptualization, Y.H.M. and R.T.; methodology, Y.H.M. and R.T.; software, Y.H.M. and R.T.; validation, R.T.; formal analysis, Y.H.M.; writing—review and editing, K.S.N., M.A., F.A., M.A.C. and S.Y.M. All authors have read and agreed to the published version of the manuscript.

**Funding:** This research received no external funding.

**Institutional Review Board Statement:** The ethical committee of (IRB & EC) of Faculty of Social Sciences and Humanities (FSSH) of Shifa Tameer-e-Millat University, has reviewed and approved. And study was conducted in accordance with the declaration of Helsinki.

**Informed Consent Statement:** Informed consent was obtained from all subjects involved in the study.

**Data Availability Statement:** Data can be provided, on request, by the corresponding author.

**Conflicts of Interest:** The authors declare no conflict of interest.

## Appendix A

**Table A1.** Confirmatory-factor analysis (factor loadings and Alpha values).

| Item Descriptions | Item No. | Loadings | Cronbach's Alpha Values |
|---|---|---|---|
| Cross-functional cooperation for environmental improvements | IEM1 | 0.933 | 0.981 |
| Special training for workers on environmental issues | IEM2 | 0.950 | |
| ISO 14000 certification | IEM3 | 0.968 | |
| Eco-labeling of products | IEM4 | 0.959 | |
| The internal performance-evaluation system incorporates environmental factors | IEM5 | 0.958 | |
| Generate environmental reports for internal evaluation | IEM6 | 0.966 | |
| Cooperation with suppliers for environmental objectives | GP1 | 0.964 | |
| Environmental audit for suppliers' inner management | GP2 | 0.971 | |
| Suppliers' ISO 14000 certification | GP3 | 0.965 | 0.979 |
| Suppliers are selected using environmental criteria | GP4 | 0.928 | |
| Provision ofdesign specifications to suppliers that include environmental requirements for purchased items | GP5 | 0.970 | |
| Cooperation with customers for cleaner production | CC1 | 0.925 | |
| Cooperation with customers to useless energy during product transportation | CC2 | 0.911 | 0.919 |
| Cooperation with customers for reverse logistical relationships | CC3 | 0.947 | |
| Design of products for reuse, recycle, recovery of material, and component parts | ECO-D1 | 0.924 | |
| Design of products to avoid or reduce use of hazardous products | ECO-D2 | 0.912 | 0.913 |
| Design of processes for minimization of waste | ECO-D3 | 0.932 | |
| Collection and recycling of end-of-life products and materials | IR1 | 0.985 | |
| Investment recovery (sale) of excess inventories/materials | IR2 | 0.985 | 0.969 |
| This firm is very concerned with environmental protection. | CSR1 | 0.954 | |
| This firm is very concerned with customers' benefits | CSR2 | 0.952 | 0.953 |
| This firm actively participates in social initiatives | CSR3 | 0.963 | |
| I regularly think about the ethical implications of my decisions | RMA1 | 0.748 | |
| I think about the morality of my actions almost every day | RMA2 | 0.837 | |
| I often find myself pondering about ethical issues | RMA3 | 0.818 | 0.856 |
| I often reflect on the moral aspects of my decisions | RMA4 | 0.797 | |
| I like to think about ethics | RMA5 | 0.779 | |
| Improved compliance with environmental standards | ENP1 | 0.968 | |
| Reduction in airborne emissions | ENP2 | 0.956 | |
| Reduction in consumption of hazardous materials | ENP3 | 0.946 | 0.962 |
| Reduction in energy consumption | ENP4 | 0.892 | |
| Reduction in material usage | ENP5 | 0.893 | |
| Decrease in costs for of purchases of materials. | ECP1 | 0.947 | |
| Decrease in cost of energy consumption | ECP2 | 0.924 | |
| Decrease in fees for waste treatment | ECP3 | 0.952 | |
| Decrease in fees for waste treatment | ECP4 | 0.943 | 0.967 |
| Decrease in fines for environmental accidents | ECP5 | 0.936 | |
| Improved overall stakeholder welfare | SCP1 | 0.903 | |
| Improvement in community health and safety | SCP2 | 0.933 | 0.955 |
| Reduction in environmental effects and risks to the general public. | SCP3 | 0.920 | |
| Improved occupational health and safety of employees | SCP4 | 0.934 | |
| Improved awareness and protection of the claims and rights of people in the community served | SCP5 | 0.912 | |
| Second-Order SP | | | |
| | ENP | 0.981 | 0.986 |
| | ECP | 0.988 | |
| | SCP | 0.981 | |

**Appendix B**

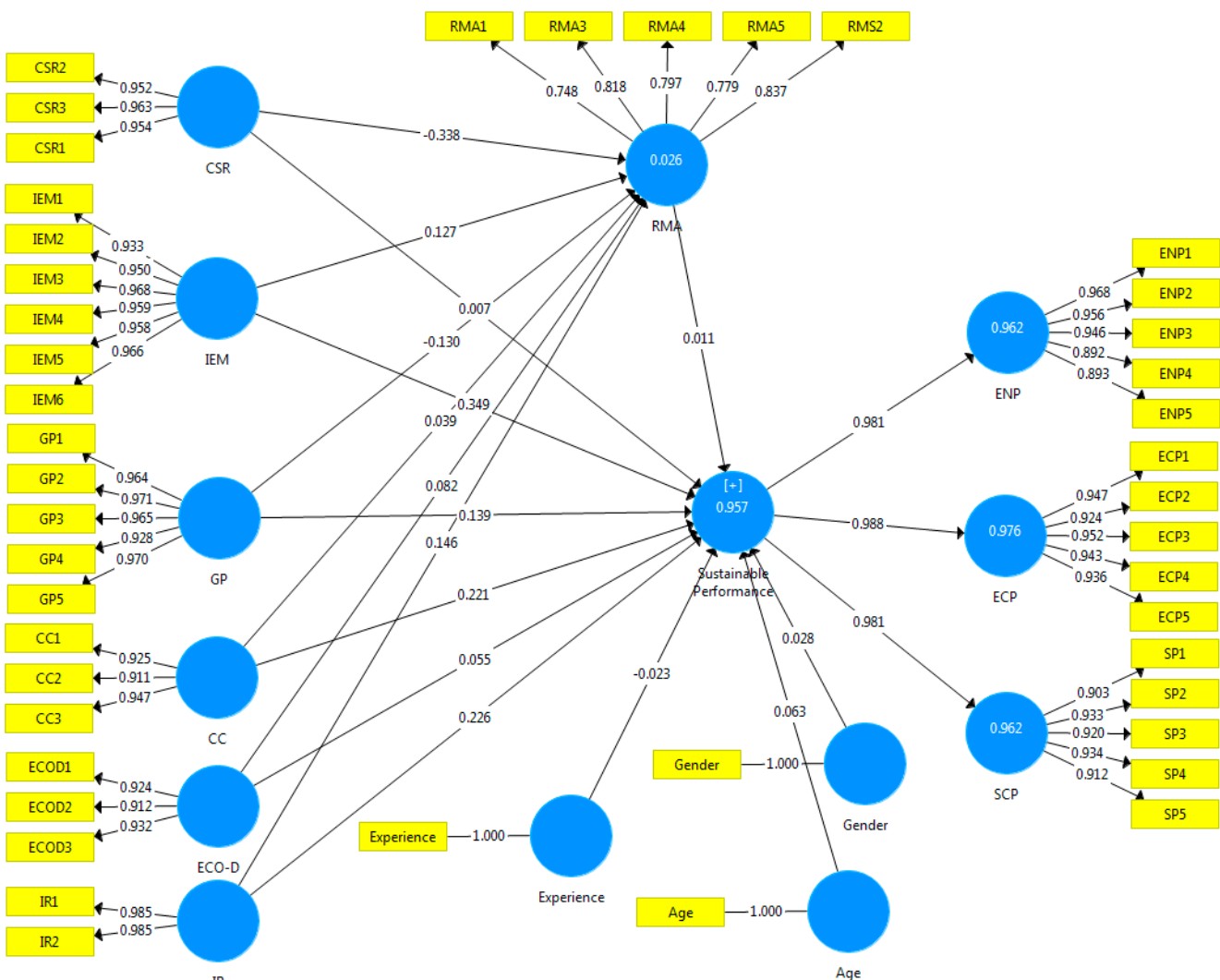

**Figure A1.** Overall best-fitting model.

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
