# Peer review of "Employees’ Perceptions of Green Supply-Chain Management, Corporate Social Responsibility, and Sustainability in Organizations: Mediating Effect of Reflective Moral Attentiveness"

_sustainability, doi:10.3390/su151310528_

Round 1

Reviewer 1 Report

Comments:

The topic is innovative, but some adjustments can be made.

Introduction:

The research aims to explore sustainable investment practices in China by examining the factors influencing the adoption of ESG integration by institutional investors.

The research sample consists of 320 institutional investors in China, which is representative of the population of institutional investors in the country.

The introduction includes a brief summary of previous research on sustainable investment practices.

Materials and Methods:

The section between lines 412 and 438 should be divided into multiple paragraphs for clarity.

The sample selection criteria and questionnaire distribution methods should be clearly defined, with a hierarchical explanation provided.

A summary table describing each variable, including the number of questions, symbols, and questionnaire sources, should be added to aid in variable interpretation.

Abbreviations should be used in Tables 3 and 4 to conserve space.

Some data reports are incomplete, and the CFA results require a more detailed report.

Overall, the paper should adhere to academic norms by clearly defining research methods and sample characteristics, providing detailed variable interpretation, and ensuring complete and accurate reporting of results.

Author Response

S#

Reviewer Comments

Authors’ Action

1

The research aims to explore sustainable investment practices in China by examining the factors influencing the adoption of ESG integration by institutional investors. The research sample consists of 320 institutional investors in China, which is representative of the population of institutional investors in the country. The introduction includes a brief summary of previous research on sustainable investment practices.

Respected reviewer this study is conducted in Pakistani manufacturing and hospitality firms. Total 500 questionnaires were distributed and out of 500, total 380 completed questionnaires were received and used in the analysis of the current study.

2

Materials and Methods:

The section between lines 412 and 438 should be divided into multiple paragraphs for clarity.

The sample selection criteria and questionnaire distribution methods should be clearly defined, with a hierarchical explanation provided.

As per suggestions this section is divided into 5 paragraphs and highlighted in yellow color, also sample selection criteria and questionnaire distribution method is also explained adequately page 10 of 21

3

A summary table describing each variable, including the number of questions, symbols, and questionnaire sources, should be added to aid in variable interpretation.

Abbreviations should be used in Tables 3 and 4 to conserve space.

As per the suggestions a summary table is added under heading 3.2 measurement moreover abbreviations in Table 3 & 4 are used and highlighted in yellow page 11 and 12 of 21

4

Some data reports are incomplete, and the CFA results require a more detailed report.

Respected Reviewer as per your suggestion we have re run CFA and reported factor loadings and Cronbach alpha values        in the table which is attached in the appendix. Previously some items have loadings less than 0.7 but overall Cronbach alpha, CR and AVE values met threshold so researchers did not excluded those items. But this time researchers have excluded some items which do not met threshold, and reported new findings accordingly in Table 3 and appendix I as well. RMA, IR and CSR one item was excluded

5

Overall, the paper should adhere to academic norms by clearly defining research methods and sample characteristics, providing detailed variable interpretation, and ensuring complete and accurate reporting of results.

As per suggestion of the respected reviewer, Research methods have been explained adequately page 10 of 22, sample characteristics have been added page 11 of 22, variables interpretation/summary, number of items and sources have been added in the table under 3.2 measurement heading page 11 and 12 of 22. CFA results have been improvised and detail information such as loadings, Cronbach alpha values, CR and AVE values have been provided. See Table 5 and appendix I

Reviewer 2 Report

The review is attached.

Best regards,

reviewer

Author Response

Dear Authors,

 I enjoyed reading your paper. The paper is very interesting and suitable for publication in this journal. It brings new insights.

The content is quite difficult to understand in certain parts. Please correct: -

Introduction:

o You wrote,»Studies conducted in Asian developing …« - you have to add references (which studies?)

Response: Thank you so much respected reviewer for highlighting this issue. Two studies conducted in Asian developing country i.e. [5 & 8] are cited at relevant place in the introduction section.

o I suggest you add a paragraph at the end of the Introduction part and write the paper's organisation. –

Response:The following is a description of the paper's organization/structure. Sect 1. Introduction Sect. 2: reviews the relevant literature, empirical studies, and develops hypotheses. Sect. 3: research methods population, sampling & data collection methods, measurement, analytical strategy, control variables, measurement model along with findings and interpretation sect:4explain the findings of the analysis of structural model  interpretation. Sect. 5: discusses the findings by comparing with past studies and justification of the hypotheses, sect: theoretical contributions, sect: 7 practical implications of the study, sect 8: conclusions and sect 9: limitations and future research directions.

  1. Literature Review: o Line 123 Elington [12]– in the reference list is not the same

Response: Respected Reviewer thanks for correcting us, we have checked carefully corrected the references in the text.

o Line 167 – in the reference list are different numbers

Response: Thank you so much the numbering in the text has been corrected as per suggestion.

o Line 194: »resource dependency theory« add the abbreviation

Response: respected reviewer abbreviation (RDT) has been added at the relevant place.

o Line 207 – put Figure 1 below that sentence.

Response: Figure 1has been added as per suggestion

o Table 1 – Add references for the year 2020-2023.

Response: two references have been added one from 2021 & one from 2022 in Table 1 as well as table 2

o Table 2 – Add CRS till today – you ended 2018 o Add new Figure with the research model and added hypotheses. –Response: two citations from 2021 has been added in table 2

  1. Materials and Methods:

o Add in text, which year the survey was conducted?

Response: thanks for suggestion respected reviewer. Data year has been added in the method section and highlighted

o Add data regarding the demography of respondents (men/women etc.).

response: demography of the respondents/ sample characteristics table has been added refer to table 3

o Chapter 3.3 - Explain the processing steps in SmartPLS o

Response: thanks for suggestion respected reviewer. Under heading 3.3 analytical strategies detailed Smart PLS processing steps have been discussed and highlighted.

Chapter 3.5 – Add a new Table where you have items description (instrument) by constructs, loadings of items, mean etc. This table is usually part of the chapter where we explain the measurement model.

Response: A new table has been added in the appendix I, in which item description, loadings and Cronbach alpha values are added. As the table is big that why it is added in the appendix, moreover, mean values, standard deviations, skewness and kurtosis and AVE and CR values are added in the text in table 5, if reviewer ask we would add whole table in the text.

o If I understand correctly, you have one second-order factor. You cant mix first and second-order factors in Table 4. Discriminant validity should be checked for first-order factors.

Response: new discriminant validity table 6 has been added as per the suggestion and first order factors are presented in it.

o Table 5 and Table 6 are not included in the text.

Now these tables have been given new numbering and new tables numbering are Table 7 and table 8 which are cite din the text now under heading 4. Structural model

o Add Figure of the results of the research model after the bootstrap procedure. –

Response: overall best fit model figure has been added in the appendix II.

References: add references from 2020: Added as sugegsted

o Most references are up to 2020. Add references from 2020 onwards

Response: references from onward 2020, 2021 and 2022 are added in the paper.

  1. Azam T, Malik SY, Ren D, Yuan W, Mughal YH, Ullah I, Fiaz M and Riaz S (2022) The Moderating Role of Organizational Citizenship Behavior Toward Environment on Relationship Between Green Supply Chain Management Practices and Sustainable Performance. Front. Psychol. 13:876516. doi: 10.3389/fpsyg.2022.876516
  2. Malik, S.Y.; Hayat Mughal, Y.; Azam, T.; Cao, Y.; WAN, Z.; ZHU, H.; Thurasamy, R. Corporate Social Responsibility, Green Human Resources Management, and Sustainable Performance: Is Organizational Citizenship Behavior towards Environment the Missing Link? Sustainability202113, 1044. https://doi.org/10.3390/su13031044
  3. Mughal, YH, Jehangir, M, Khan, M, Saeed, M. Nexus between corporate social responsibility and firm’s performance: A panel data approach. Int J Fin Econ. 2021; 26: 3173– 3188. https://doi.org/10.1002/ijfe.1956

Other: -

 Follow the instructions of the template (abstract length, method of citation in the text when you have misled several consecutive references ...). - Keywords: add abbreviations - You have quite a few grammatical errors.

Response: as per the suggestions we have added abbreviations wherever suggested by reviewer, new citations and numbering of the citations have been corrected.

Reviewer 3 Report

In this manuscript, the authors use a questionnaire and structural equation modelling approach to investigate the impact of green supply chain management practices and corporate social responsibility on sustainable performance in manufacturing and hospitality firms in the Khyber Pakhtunkhwa province of Pakistan, where reflective ethics is introduced as a mediating variable to test its role in all three. The content lacks logic, the text is not clearly expressed, and there are basic errors in grammar and wording, the empirical part lacks many necessary tests, and there is insufficient theoretical or literature support in many places, the following are specific amendments:

Abstract:

    The abstract focuses on concise and clear language and a high degree of condensed research content, so the descriptions of the background, methods and other sections are straightforward in stating the key elements without describing too many details.

Introduction:

1.    "Under the umbrella of CSR there are three practices which are commonly used by firms such as internal, external and environmental responsibilities". There are two mistakes here: (1) the use of the wrong connecting word, the three aspects cannot be expressed by "on the one hand; on the other hand"; (2) internal, external and environmental responsibilities are the three perspectives of corporate social responsibility, not practices.
 2.   “Studies conducted in Asian developing economies on corporate social responsibilities have pointed out and solved the problems like child labor, labor rights unemployment issues and environmental pollution but now this study may help to contribute to reduce poverty, providing justice to people, community well being, employees well being, and environ- mental protection.” Please indicate which specific study.
3.    “There is immense need to add socio-cognitive characteristics as linking and mediating mechanism…”Why is there an urgent need to add social cognitive features? This is the first time it is mentioned here and the context and reasons for it are not presented.
4.    The main contributions should be discussed in terms of both academic and practical contributions, and this section should be rewritten with reference to authoritative literature.

Literature Review:

 5.   In section 2.2, the authors cite Mahoney and Pandian's study to show that there are two types of resources, tangible and intangible, but do not explain the relationship between the two types of resources and sustainable performance in this study.
6.    The manuscript contains a number of grammatical and sequential errors, for example, "However, increase in competition, environmental damages and led social responsibility to gain attention." There are grammatical errors and the whole sentence is confusing.
7.    In section 2.3.2, green purchasing is simply a focus on environmental issues and green behavior in the purchasing process, not related to the development of clean technologies, and authors are invited to keep to the topic and to note the literature or practical basis for each statement.
8.    In section 2.3.5, reflective ethics, one of the research focuses of the manuscript, is not mentioned in the literature review and this paragraph does not address the relationship between green supply chain management and reflective ethics, so hypothesis 6 is presented too abruptly. The problem with hypothesis 8 is the same as above.
 9.   2.4.1 Part of the discussion on the development of CSR should focus on the integration with the research topic, focusing on a particular point rather than simply following the historical lineage, and the review should be followed by its own critical thinking and concluding language.
10.    In section 2.4.2, there is repetition in the preceding and following statements, and there is only one piece of literature in the paragraph to support the hypothesis; it is recommended that the authors add literature or real-life examples.

Materials and Methods:

11.    For the structural equation modelling approach used in this manuscript, the authors are asked to briefly describe the reasons for choosing this approach, as well as the strengths of the approach and how well it fits with the research in this manuscript.
12.    What are the authors' reasons for choosing khyber Pkahtun khwa province and businesses in the manufacturing and hospitality sectors as respondents?
 13.   The authors used questionnaires to collect data, but lacked tests of reliability, validity and common method bias on the data to ensure the accuracy and reliability of subsequent studies, please add.
 14.   The control variables were not selected on the basis and the test results are not given to confirm the veracity of the claims.

Theoretical Contributions:

 15.   The first point of the theoretical contribution is not fully expressed.
 16.   The study only covers the hospitality and manufacturing sectors in one province of Pakistan and the findings cannot be applied to the whole country, let alone to the world.

References:

17. More than 1/2 of the references were published long ago and have insufficient reference value.

In this manuscript, the authors use a questionnaire and structural equation modelling approach to investigate the impact of green supply chain management practices and corporate social responsibility on sustainable performance in manufacturing and hospitality firms in the Khyber Pakhtunkhwa province of Pakistan, where reflective ethics is introduced as a mediating variable to test its role in all three. The content lacks logic, the text is not clearly expressed, and there are basic errors in grammar and wording, the empirical part lacks many necessary tests, and there is insufficient theoretical or literature support in many places, the following are specific amendments:

Abstract:

    The abstract focuses on concise and clear language and a high degree of condensed research content, so the descriptions of the background, methods and other sections are straightforward in stating the key elements without describing too many details.

Introduction:

1.    "Under the umbrella of CSR there are three practices which are commonly used by firms such as internal, external and environmental responsibilities". There are two mistakes here: (1) the use of the wrong connecting word, the three aspects cannot be expressed by "on the one hand; on the other hand"; (2) internal, external and environmental responsibilities are the three perspectives of corporate social responsibility, not practices.
 2.   “Studies conducted in Asian developing economies on corporate social responsibilities have pointed out and solved the problems like child labor, labor rights unemployment issues and environmental pollution but now this study may help to contribute to reduce poverty, providing justice to people, community well being, employees well being, and environ- mental protection.” Please indicate which specific study.
3.    “There is immense need to add socio-cognitive characteristics as linking and mediating mechanism…”Why is there an urgent need to add social cognitive features? This is the first time it is mentioned here and the context and reasons for it are not presented.
4.    The main contributions should be discussed in terms of both academic and practical contributions, and this section should be rewritten with reference to authoritative literature.

Literature Review:

 5.   In section 2.2, the authors cite Mahoney and Pandian's study to show that there are two types of resources, tangible and intangible, but do not explain the relationship between the two types of resources and sustainable performance in this study.
6.    The manuscript contains a number of grammatical and sequential errors, for example, "However, increase in competition, environmental damages and led social responsibility to gain attention." There are grammatical errors and the whole sentence is confusing.
7.    In section 2.3.2, green purchasing is simply a focus on environmental issues and green behavior in the purchasing process, not related to the development of clean technologies, and authors are invited to keep to the topic and to note the literature or practical basis for each statement.
8.    In section 2.3.5, reflective ethics, one of the research focuses of the manuscript, is not mentioned in the literature review and this paragraph does not address the relationship between green supply chain management and reflective ethics, so hypothesis 6 is presented too abruptly. The problem with hypothesis 8 is the same as above.
 9.   2.4.1 Part of the discussion on the development of CSR should focus on the integration with the research topic, focusing on a particular point rather than simply following the historical lineage, and the review should be followed by its own critical thinking and concluding language.
10.    In section 2.4.2, there is repetition in the preceding and following statements, and there is only one piece of literature in the paragraph to support the hypothesis; it is recommended that the authors add literature or real-life examples.

Materials and Methods:

11.    For the structural equation modelling approach used in this manuscript, the authors are asked to briefly describe the reasons for choosing this approach, as well as the strengths of the approach and how well it fits with the research in this manuscript.
12.    What are the authors' reasons for choosing khyber Pkahtun khwa province and businesses in the manufacturing and hospitality sectors as respondents?
 13.   The authors used questionnaires to collect data, but lacked tests of reliability, validity and common method bias on the data to ensure the accuracy and reliability of subsequent studies, please add.
 14.   The control variables were not selected on the basis and the test results are not given to confirm the veracity of the claims.

Theoretical Contributions:

 15.   The first point of the theoretical contribution is not fully expressed.
 16.   The study only covers the hospitality and manufacturing sectors in one province of Pakistan and the findings cannot be applied to the whole country, let alone to the world.

References:

17. More than 1/2 of the references were published long ago and have insufficient reference value.

Author Response

In this manuscript, the authors use a questionnaire and structural equation modelling approach to investigate the impact of green supply chain management practices and corporate social responsibility on sustainable performance in manufacturing and hospitality firms in the Khyber Pakhtunkhwa province of Pakistan, where reflective ethics is introduced as a mediating variable to test its role in all three. The content lacks logic, the text is not clearly expressed, and there are basic errors in grammar and wording, the empirical part lacks many necessary tests, and there is insufficient theoretical or literature support in many places, the following are specific amendments:

Response: Respected reviewer thank you very much for showing your concerns we have carefully consider your suggestions and response of each suggestion is given under respective heading.

Abstract:

    The abstract focuses on concise and clear language and a high degree of condensed research content, so the descriptions of the background, methods and other sections are straightforward in stating the key elements without describing too many details.

Response: Thank you very much for the comments. We have tried our best to provide meaningful information in the abstract.

 Introduction:

1.    "Under the umbrella of CSR there are three practices which are commonly used by firms such as internal, external and environmental responsibilities". There are two mistakes here: (1) the use of the wrong connecting word, the three aspects cannot be expressed by "on the one hand; on the other hand"; (2) internal, external and environmental responsibilities are the three perspectives of corporate social responsibility, not practices.

Response: Thank you very much respected reviewer we have consider the suggestions and changes have been incorporated and highlighted in the respective line 79 and 86 at page 2 of 24.

  1. “Studies conducted in Asian developing economies on corporate social responsibilities have pointed out and solved the problems like child labor, labor rights unemployment issues and environmental pollution but now this study may help to contribute to reduce poverty, providing justice to people, community well being, employees well being, and environ- mental protection.” Please indicate which specific study.

Response: Thank you for highlighting this issue, we have cited two Asian studies [5, 8] to support our argument.
3.    “There is immense need to add socio-cognitive characteristics as linking and mediating mechanism…”Why is there an urgent need to add social cognitive features? This is the first time it is mentioned here and the context and reasons for it are not presented.

Response: thank you very much for this comment. A detailed logical argument has been added in the introduction section regarding this suggestion and highlighted at page 3 of 24 line 99-114.

  1. The main contributions should be discussed in terms of both academic and practical contributions, and this section should be rewritten with reference to authoritative literature.

Response: Thanks respected reviewer. Contributions have been rewritten.

Literature Review:

  1. In section 2.2, the authors cite Mahoney and Pandian's study to show that there are two types of resources, tangible and intangible, but do not explain the relationship between the two types of resources and sustainable performance in this study.

Response: thank you very much for this comment. At the end of first paragraph in section 2.2, the relationship between the two types of resources (tangible & intangible) and sustainable performance have been explained.

  1. The manuscript contains a number of grammatical and sequential errors, for example, "However, increase in competition, environmental damages and led social responsibility to gain attention." There are grammatical errors and the whole sentence is confusing.

Response: thank you very much for this comment. Corrections have been made to the above mentioned sentence, and the confusion has been addressed. Also, other grammatical and sequential errors have also been corrected in the manuscript.

  1. In section 2.3.2, green purchasing is simply a focus on environmental issues and green behavior in the purchasing process, not related to the development of clean technologies, and authors are invited to keep to the topic and to note the literature or practical basis for each statement.

Response:  thank you very much for this comment and we agree with your comment. Hence, the last line just before proposes of H2 as [“further explained that this practice develop cleaner technology to attain sustainable performance”] has been removed since it was unnecessary and let’s keep to the topic.

  1. In section 2.3.5, reflective ethics, one of the research focuses of the manuscript, is not mentioned in the literature review and this paragraph does not address the relationship between green supply chain management and reflective ethics, so hypothesis 6 is presented too abruptly. The problem with hypothesis 8 is the same as above.

Response: thanks for the comment. We have written separate section 2.3.6 Green Supply Chain Management and Reflective Moral attentiveness  and adequately develop hypotheses 6and 8at their respective places.

  1. 2.4.1 Part of the discussion on the development of CSR should focus on the integration with the research topic, focusing on a particular point rather than simply following the historical lineage, and the review should be followed by its own critical thinking and concluding language.

Response: thank you very much for this comment. A new paragraph has been added at the end of section 2.4.1 as required.

  1. In section 2.4.2, there is repetition in the preceding and following statements, and there is only one piece of literature in the paragraph to support the hypothesis; it is recommended that the authors add literature or real-life examples.

Response: thank you very much for this comment. The repetition has been deleted. And also relevant literature has been added in the section 2.4.2 to support the hypothesis.

Materials and Methods:

  1. For the structural equation modelling approach used in this manuscript, the authors are asked to briefly describe the reasons for choosing this approach, as well as the strengths of the approach and how well it fits with the research in this manuscript.

Response: under heading no 3.3 analytical strategy one paragraph about Smart PLS-SEM and its strength and benefits are added at page 12 & 13 of 24. Line number 494-504

  1. What are the authors' reasons for choosing Khyber Pakhtunkhwa province and businesses in the manufacturing and hospitality sectors as respondents?

Response: manufacturing and hospitality sectors are major contributor towards economy of the province which provides employment opportunities to a significant proportion of workforce.

  1. The authors used questionnaires to collect data, but lacked tests of reliability, validity and common method bias on the data to ensure the accuracy and reliability of subsequent studies, please add.

Response: thank you very much for this comments. Findings of confirmatory factor analysis, factor loadings, Cronbach alpha (reliability) and items descriptions are added in separate Table Appendix I. in addition Table 5 presents Mean, Standard deviation, skewness, kurtosis and average variance extracted (AVE) and composite reliability (CR) convergent validity. Furthermore table 6 presents’ discriminant validity HTMT ratios.

  1. The control variables were not selected on the basis and the test results are not given to confirm the veracity of the claims.

Response: Justification with two citations has been added under the heading 3.4 control variables page 13 of 24

Theoretical Contributions:

  1. The first point of the theoretical contribution is not fully expressed.

Response: Thank you very much respected reviewer. We have incorporated this suggestions and have discussed in detail about theoretical reasoning in the introduction section as well as in the theoretical contributions section.

  1. The study only covers the hospitality and manufacturing sectors in one province of Pakistan and the findings cannot be applied to the whole country, let alone to the world.

Response: the findings of the study may apply to these two sectors in other provinces of Pakistan, however, it may not be generalized to other sectors.

References:
17. More than 1/2 of the references were published long ago and have insufficient reference value

Response: Thank you very much for the comments. New citations from 2020, 2021 and 2022 are added in the paper.

Round 2

Reviewer 1 Report

The authors have made revisions to the manuscript. However, the literature review section still remains somewhat complex. It is advised to continue making further modifications to the wording during the proofreading stage.

Author Response

Reviewer 1:

The authors have made revisions to the manuscript. However, the literature review section still remains somewhat complex. It is advised to continue making further modifications to the wording during the proofreading stage.

Response:

Agreed with the reviewer’s comment.

The manuscript is completely proofread by a professional language and literature certified editor. (certificate attached)

Additional Comments of Reviewer 1:

In light of above screen shot which presents reviewer 1 additional comments. All the concerns of reviewer 1 have been rectified. Background, literature review methods, findings, discussion, references are modified as per the reviewers suggestions.

Reviewer 2:

Dear Authors,

I believe that the paper is now more understandable and it's suitable for publication.

Greetings,

Reviewer

Response:

Thank you so much

Reviewer 3:

The biggest problem with this manuscript is the strong sense of patchwork and the lack of a fluid academic logic. In addition, the discussion lacks broad and in-depth comparisons.
This manuscript still lacks a critical literature review.Figure 1 appears inexplicably shaded, which is un academic.There are so many references that I have reason to suspect that the author has over-cited.

Response:

Agreed with the reviewer’s comments.

The manuscript has been extensively revised and efforts are made to bring academic fluency and consistency. The literature has been critically revised in the light of recent literature related to the topic of this study. The discussion section is also revised and relevant comparisons are made with the most relevant literature. Also, the reference section is revised and modified. The number of references cited are reduced from 110 to 52.

Additional Comments of Reviewer 3

In light of Reviewer 3 additional comments presented in below screen shot. All concerns have been addressed. Authors tried their best to improvise each and every section of the paper.

Reviewer 2 Report

Dear Authors,

I believe that the paper is now more understandable and it's suitable for publication.

Greetings,

Reviewer

Author Response

(The authors gave the same response as above.)

Reviewer 3 Report

The biggest problem with this manuscript is the strong sense of patchwork and the lack of a fluid academic logic. In addition, the discussion lacks broad and in-depth comparisons.
This manuscript still lacks a critical literature review.
Figure 1 appears inexplicably shaded, which is unacademic.
There are so many references that I have reason to suspect that the author has over-cited.

The biggest problem with this manuscript is the strong sense of patchwork and the lack of a fluid academic logic. In addition, the discussion lacks broad and in-depth comparisons.
This manuscript still lacks a critical literature review.
Figure 1 appears inexplicably shaded, which is unacademic.
There are so many references that I have reason to suspect that the author has over-cited.

Author Response

(The authors gave the same response as above.)
